# Reprogramming and Differentiation of Cutaneous Squamous Cell Carcinoma Cells in Recessive Dystrophic Epidermolysis Bullosa

**DOI:** 10.3390/ijms22010245

**Published:** 2020-12-29

**Authors:** Avina Rami, Łukasz Łaczmański, Jagoda Jacków-Nowicka, Joanna Jacków

**Affiliations:** 1Department of Dermatology, Columbia University, New York, NY 10032, USA; ar3867@cumc.columbia.edu; 2Laboratory of Genomics &Bioinformatics, Hirszfeld Institute of Immunology and Experimental Therapy, Polish Academy of Sciences, 53-114 Wroclaw, Poland; lukasz.laczmanski@hirszfeld.pl; 3Department of General and Interventional Radiology and Neuroradiology, Wroclaw Medical University, 50-556 Wroclaw, Poland; jagoda.jackow-nowicka@umed.wroc.pl; 4St John’s Institute of Dermatology, King’s College London, Guy’s Campus, London SE1 9RT, UK

**Keywords:** induced pluripotent stem cell technology, squamous cell carcinoma, recessive dystrophic epidermolysis bullosa, cancer, biomarkers, in vitro model

## Abstract

The early onset and rapid progression of cutaneous squamous cell carcinoma (cSCC) leads to high mortality rates in individuals with recessive dystrophic epidermolysis bullosa (RDEB). Currently, the molecular mechanisms underlying cSCC development in RDEB are not well understood and there are limited therapeutic options. RDEB-cSCC arises through the accumulation of genetic mutations; however, previous work analyzing gene expression profiles have not been able to explain its aggressive nature. Therefore, we generated a model to study RDEB-cSCC development using cellular reprograming and re-differentiation technology. We compared RDEB-cSCC to cSCC that were first reprogrammed into induced pluripotent stem cells (RDEB-cSCC-iPSC) and then differentiated back to keratinocytes (RDEB-cSCC-iKC). The RDEB-cSCC-iKC cell population had reduced proliferative capacities in vitro and in vivo, suggesting that reprogramming and re-differentiation leads to functional changes. Finally, we performed RNA-seq analysis for RDEB-cSCC, RDEB-cSCC-iPSC, and RDEB-cSCC-iKC and identified different gene expression signatures between these cell populations. Taken together, this cell culture model offers a valuable tool to study cSCC and provides a novel way to identify potential therapeutic targets for RDEB-cSCC.

## 1. Introduction

Dystrophic epidermolysis bullosa (DEB) is caused by mutations in the *COL7A1* gene, which impair the function of type VII collagen (C7), resulting in blister formation, altered wound-healing processes, and the development of highly aggressive cutaneous squamous cell carcinoma (cSCC) that may lead to metastasis. cSCC, a malignant growth of the cutaneous epithelium, is the second most common skin cancer. In particular, patients with severe recessive DEB (RDEB) have a high risk of early death due to metastasis from their cSCC. This risk reaches up to 87% by the age of 45 years [1,2].

The mechanisms underlying RDEB-cSCC tumorigenesis are not fully understood and there are limited current options to treat the disease. Therefore, understanding what drives cSCC in RDEB is crucial to prevent or halt cancer progression in patients. Several mechanisms are thought to give rise to the aggressive and quick-advancing nature of RDEB-cSCCs [3]. Recurrent wounding and blistering due to loss of type VII collagen (C7) promotes the production of pro-matrix metalloproteinase 2, epithelial-mesenchymal transition, and increased TGF-b signaling with microbial challenges, all of which cause a pro-inflammatory microenvironment and are considered major risk factors [4,5,6,7,8]. To date, mutation signatures and transcriptomic analyses have been previously characterized in RDEB skin, RDEB-cSCC tumors and isolated RDEB fibroblasts [4,9,10].

Although primary cells can be used to study genetic changes, these cells are limited in long-term studies due to their relatively short lifespan before senescence [11]. The development of iPSC from primary cells allows for long-term cell expansion and retention of an undifferentiated state, allowing for a scalable source of material for lasting studies. Somatic cells can be reprogrammed to manifest stem-cell-like morphology and characteristics. Four factors (*OCT4*, *SOX2*, *KLF4* and *NANOG*) are sufficient to reprogram somatic cells to induced pluripotent stem cells (iPSC) that resemble the characteristics and morphology of embryonic stem cells [12]. Reprogramming of cancer cells has been previously reported for skin cancer, melanoma, gastric cancer, and leukemia [13,14,15,16]. Common features of reprogrammed cancer cells (cancer-iPSC) include the expression of various pluripotency markers, teratoma formation, and induced dedifferentiation.

Cellular reprogramming technology can be used to further understand epigenetics-driven cancer development [17]. Previous papers have highlighted how reprogramming technology can be used to modify the epigenome without affecting the underlying genomic sequences to study the onset of cancer [18]. Thus, in our study, we hypothesized that using our reprogramming and re-differentiation model, we can study epigenetic alterations without modifying the underlying genomic sequences.

Here, we used iPSC reprogramming technology to develop a model that can be used to study RDEB-cSCC tumorigenesis. Specifically, we reprogrammed parental cSCC cells derived from patients with RDEB through the introduction of the four Yamanaka factors (OKSM) using episomal plasmids to create RDEB-cSCC-induced pluripotent stem cells (RDEB-cSCC-iPSC). Furthermore, we differentiated the cSCC-iPSC into induced keratinocytes (RDEB-cSCC-iKC) to recapitulate the changes that may occur during the early stages of cancer. The reprogramming and differentiation of RDEB-cSCC may represent an earlier stage in the evolution of aggressive cSCC. Further, RNA-seq analysis was performed to identify potential biomarkers and therapeutic targets for RDEB-cSCC tumorigenesis. Taken together, the reprogramming and differentiation of cancer cells creates a powerful tool for distinguishing genetic alterations that occur during cSCC tumor initiation and progression.

## 2. Results

### 2.1. Reprogramming of RDEB-cSCC into RDEB-cSCC-iPSC

To elucidate the possibility of reprogramming human RDEB-cSCC into RDEB-cSCC-iPSC, an epigenetically distinct cell population, we tested two reprogramming methods on the RDEB-cSCC lines [19]. We first attempted to reprogram RDEB-cSCC into RDEB-cSCC-iPSC using a lentivirus-based reprogramming method (Figure 1A). Following this attempt, we then tested feeder-free reprogramming methods of RDEB-cSCC into RDEB-cSCC-iPSCs using an episomal plasmid (Figure 1B). We decided to carry on with the second method since the morphology of the RDEB-SCC-iPSCs derived from RDEB-SCCs were compared to iPSCs derived from fibroblasts (RDEB-FB) and appeared to be morphologically similar (small cell shape, large nuclear-to-cytoplasmic ratio, prominent nucleoli, tight and flat colonies) (Figure 1C). The pluripotency signatures in RDEB-cSCC-iPSC derived from RDEB-cSCC and RDEB-iPSCs derived from RDEB-FB were confirmed by immunofluorescence staining of four stem cell markers including SOX2, NANOG, TRA-1-60 and TRA1-81 (Figure 1D).

### 2.2. Re-Differentiation of RDEB-cSCC-iPSCs into RDEB-cSCC-iKCs

We next evaluated whether the RDEB-cSCC-iPSCs could be re-differentiated to RDEB-cSCC keratinocytes (RDEB-cSCC-iKC) to analyze the in vitro tumorigenicity potential of RDEB-cSCCs compared to RDEB-cSCC-iKC. We performed the in vitro differentiation of RDEB-cSCC-iPSCs into RDEB-cSCC-iKC using a differentiation protocol to generate iKC from iPSCs. On the basis of our recent studies, we used daily administration of 1 uM retinoic acid and 10 ng/uL bone morphogenetic protein 4 (BMP4) for 6 days when differentiating RDEB-cSCC-iPSCs into RDEB-cSCC-iKC [20]. The schematic presentation of the differentiation strategy used for generating RDEB-cSCC-iKCs from RDEB-cSCCs is shown in Figure 2A,B. Briefly, the cells were kept for maturation up to 45 days. Then, the RDEB-cSCC-iKC were expanded and prepared for further functional studies. Representative pictures of the directed differentiation of RDEB-cSCC-iPSC into RDEB-cSCCs-iKCs are shown in Figure 2C, highlighting embryoid body formation and changes in cell morphology throughout the differentiation process.

### 2.3. In Vitro Tumorigenicity of Reprogrammed and Re-Differentiated RDEB-cSCC-iKCs

The reprogrammed and re-differentiated RDEB-cSCC-iKC differ in morphology compared to the original RDEB-cSCC (Figure 3A). Based on these results, we next asked whether the in vitro tumorigenic potential of RDEB-cSCC decreased after reprogramming. For this reason, we evaluated the proliferative capacity of the newly established RDEB-cSCC-iKC cell population by a colorimetric method to determine the number of viable cells in proliferation. Interestingly, the proliferation of RDEB-cSCC-iKC measured over time (24 h, 48 h and 72 h) was significantly lower compared to their parental RDEB-cSCC cells (Figure 3B). We concluded from this experiment that the reprogramming and re-differentiation processes significantly reduced the cell’s proliferative capacity, suggesting that this method leads to the reduction of tumorigenic potential of RDEB-cSCC-iKC compared to RDEB-cSCCs.

### 2.4. In Vivo Tumorigenicity of Reprogrammed and Re-Differentiated RDEB-cSCC-iKCs

Next, we assessed the in vivo tumorigenicity of the reprogrammed and re-differentiated RDEB-cSCC-iKCs. We used our established xenotransplantation tumor model and generated 3D tumor skin constructs with RDEB-cSCC and RDEB-cSCC-iKCs expressing the luciferase gene, which were then grafted onto immunodeficient mice as previously described [21]. We used 1000 RDEB-cSCC and RDEB-cSCC-iKCs, which were spiked into normal human keratinocytes to generate the 3D skin constructs. Tumors were formed in mice grafted with parental RDEB-cSCC and RDEB-cSCC-iKCs within the course of four weeks (Figure 3C). The tumors formed by parental cells grown faster and were bigger compared to the tumors generated from RDEB-cSCC-iKCs, as shown by the bioluminescence signal intensities.

### 2.5. RDEB-cSCC and RDEB-cSCC-iKCs are Transcriptionally Different Cells

Based on the morphological and functional differences between RDEB-cSCC and RDEB-cSCC-iKCs, we initialized RNA-seq analysis to compare gene expression profiles between these samples (Figure 4). First, we analyzed expression in RDEB-cSCC and RDEB-cSCC-iPSCs. Differential expression analyses showed two clusters including genes with two opposite expression levels in RDEB-cSCC and RDEB-cSCC-iPSC cells (Figure 5A). The first cluster includes genes with higher expression, while the second includes genes with lower expression in RDEB-cSCC-iPSC cells. The ToppGene tool was used to prioritize and rank candidate genes based on functional similarity to the training gene list. The result showed that seven genes in cluster 1 (*MALL*, *STAT1*, *TNFAIP3*, *SPP1*, *MAGEA4*, *LRATD2*, *CTNNB1*) and three genes in cluster 2 (*ACACB*, *TCF7L2*, *BHLHE23*) are potential RDEB-cSCC-iPSCs differentiation markers (Figure 5B). The output of the Database for Annotation, Visualization, and Integrated Discovery (DAVID) bioinformatics tool revealed that three genes from cluster 1 (*CTNNB1*, *SPP1*, *STAT1*) are members of one metabolic pathway–perinuclear region of cytoplasm (GO:0048471) and no genes from cluster 2. Then we analyzed the expression in RDEB-cSCC and RDEB-cSCC-iKCs. Differential expression analyses showed two clusters including genes with two opposite expression levels in RDEB-cSCC and RDEB-cSCC-iKC cells (Figure 5C). The first cluster includes genes with higher expression, while the second includes genes with lower expression in RDEB-cSCC-iKC cells. The ToppGene tool analysis showed that five genes in cluster 1 (*SOD2*, *KLF12*, *DNAJC12*, *IFIH1*, *CHI3L1*) and five genes in cluster 2 (*SCEL*, *FEN1*, *SULF1*, *MCM5*, *UNG*) are potential RDEB-cSCC-iKC differentiation markers (Figure 5D). The output of the DAVID bioinformatics tool revealed that three genes from cluster 1 (*KLF12*, *IFIH1*, *SOD2*) are members of the DNA binding metabolic pathway (GO:0003677) and four genes from cluster 2 (*CDC45*, *DTL*, *FEN1*, *MCM5*) are members of the DNA replication pathway (GO:0006260).

## 3. Discussion

In this study, we have used iPSC technology to develop a novel cell culture model to recapitulate the changes that occur during the early stages of RDEB-cSCC. This reprogramming and re-differentiation method can be used as a unique tool to study cancer behavior and model disease pathogenesis. This model, now established for RDEB-cSCC, can be translated and further applied to different types of carcinomas. Through this study, we have also shown that the reprogramming and re-differentiation of RDEB-cSCC cells into induced keratinocytes facilitated functional and transcriptional changes in the transformed RDEB-cSCC-iKCs. To study mutational changes requires the ability to follow molecular alterations retrospectively. Recent progress in iPSC reprogramming and re-differentiation techniques has been instrumental in accomplishing this goal, as such techniques allow cells to obtain transcriptionally different cell signatures. One study by Kim et al. highlights how the reprogramming of somatic cells reverses the process of cell specification through epigenetic modifications, allowing for the removal of tissue-specific DNA methylation and the re-establishment of embryonic methylation patterns [22]. Additionally, other studies have underscored the use of iPSC models to study early changes in cancer. These studies have demonstrated the value of using reprogramming technology to model disease pathogenesis for colon cancer, osteosarcoma, and gastrointestinal cancer [15,23,24].

These transcriptional alterations can be further studied to understand the changes that occur during the reprogramming process. Through comparative gene expression analyses between the parental and transformed cell lines, we have uncovered iKC differentiation markers and potential biomarkers to study cSCC tumorigenesis. Taken together, this cell culture model provides a unique tool to gain insight into the pathomechanism of cSCC development in RDEB patients.

Here, we demonstrated the ability to reprogram RDEB-cSCC into RDEB-cSCC-iPSC. As previous literature has reported, several factors can influence the reprogramming of cancer cells into cancer-iPSCs. Biological obstacles such as genetic mutations, epigenetic alterations, the collection of DNA damage, or reprogramming-induced cellular senescence may affect the reprogramming efficiency of cancer cell lines [15,25]. For example, epigenetic changes during the process of reprogramming can lead to the reduced expression of pluripotency genes [26]. Moreover, oncogenic mutations present in cancer-iPSCs can lead to a reduced degree of maturation during cellular differentiation, highlighting potential genetic barriers to efficient reprogramming [16]. However, although such barriers exist, the reprogramming of cancer cells into cancer-iPSCs remains a unique tool for modeling disease pathogenesis.

After reprogramming the parental cell line to iPSC, we re-differentiated RDEB-cSCC-iPSC into RDEB-cSCC-iKC to further understand the changes that may occur during the initial stages of cancer. iKC differentiation is a technique that has been well-established and used for various gene and cell therapies. Previous papers have cited the generation of iKC from iPSC derived from the fibroblasts of RDEB patients for the therapeutic development of stem-cell-based treatments [20,27]. Here, we have shown for the first time a new utility of the iKC differentiation protocol. The differentiation of cancer-iPSC opens a new avenue to study retrospective changes in parental cancer cells. Through this method, our group has been able to study the alterations that occur during cancer initiation and progression, as well as identify potential biomarkers that may be relevant for tumor development. Recently, a study published by Ji et al. identified four tumor subpopulations within cSCC, with one population composed of tumor-specific keratinocytes unique to cancer [28]. A comparison of differential expression analyses between tumor-specific keratinocytes and RDEB-cSCC-iKC may provide more insight on the utility and validity of the iKC model.

Global reprogramming and re-differentiation of RDEB-cSCC into iKC changed the functional properties of the cells, both in vitro and in vivo. To test the in vitro tumorigenic potential of the cell lines, we performed a 3-(4,5-dimethylthiazol-2-yl)-5-(3-carboxymethoxyphenyl)-2-(4-sulfophenyl)-2H-tetrazolium) (MTS) assay to determine their proliferative capacities. Following the proliferation assay, we discovered that the RDEB-cSCC-iKC had a reduced proliferative capacity compared to RDEB-cSCC, highlighting the reduction in in vitro tumorigenic potential. To study the in vivo tumorigenic potential, we created 3D tumor skin constructs from RDEB-cSCC and RDEB-cSCC-iKC, which were then grafted onto immunodeficient mice. From these tumor models, we found the tumors formed by RDEB-cSCC-iKC grew slower and were smaller than those formed by RDEB-cSCC, suggesting a reduction in in vivo tumorigenic potential. In all, these studies demonstrate that the reprogramming and re-differentiation of RDEB-cSCC into iKC reduces overall proliferative capacity and tumorigenic potential, suggesting that cellular reprogramming and re-differentiation leads to functional changes in transformed cells.

Comprehensive profiling of the somatic alterations and transcriptomes has been widely studied in different contexts of RDEB and RDEB-cSCC. A recent study by Cho et al. has found that after sequencing multiple RDEB-SCC tumors, the carcinomas were driven by APOBEC-associated mutagenesis, elucidating a new therapeutic target for RDEB-SCC patients [4]. Additionally, another review has highlighted how mutations in *TP53*, *NOTCH1*, *NOTCH2*, *CDKN2A*, *HRAS*, and *FAT1* are considered potential drivers of RDEB-cSCC tumorigenesis [29]. Taken together, these studies demonstrate the role of sequencing and transcriptome analyses in determining new potential biomarkers and novel candidates for therapeutic targets. In further studies, we plan to examine the epigenetic signatures of the parental, reprogrammed, and re-differentiated cells. Though epigenetic changes and genomic mutations are not independent of one another, we believe studying epigenetic regulation in the iKC with cSCC mutations is useful as epigenetic changes have been previously reported to drive gene expression. A recent study by Atlasi et al. highlighted how differential gene expression can be driven by enhancer activation and chromatin accessibility, delineating the role of epigenetic modulation in driving changes in gene expression, independent of mutational changes [30]. Additionally, studying epigenetic changes can be useful when developing targeted therapies for cancer. A study by Miyoshi et al. demonstrated how tumor-suppressor gene *P16* was repressed in induced pluripotent cancer cells, highlighting a potential target for gastrointestinal cancer treatment [15].

Our results uncovered that five genes from cluster 1 (*SOD2*, *KLF12*, *DNAJC12*, *IFIH1*, *CHI3L1*) and five from cluster 2 (*SCEL*, *FEN1*, *SULF1*, *MCM5*, *UNG*) were potential RDEB-cSCC-iKC differentiation markers. SOD2 is an enzyme involved in the clearance of reactive oxygen species that is up-regulated in RDEB-cSCC-iKC compared to the parental RDEB-cSCC cell line (*p* < 0.05). The loss of SOD2 expression has been reported to occur early in tumor progression, allowing the propagation of aggressive tumors and metastasis following the increase in free radical production [31]. Moreover, FEN1 is a protein involved in DNA repair that is down-regulated in RDEB-cSCC-iKC compared to the parental RDEB-cSCC cell line (*p* < 0.05). The down-regulation of the *FEN1* gene has been implicated in causing genomic instability and cancer predisposition [32]. These results suggest a potential reason behind the increase in mutations for RDEB patients, highlighting possible issues associated with DNA repair mechanisms. In sum, these genes, as well as others within the clusters, serve as plausible biomarkers for tumorigenesis and cancer susceptibility, as well as possible therapeutic targets for cSCC.

Altogether, we have shown that reprogramming and re-differentiation of parental RDEB-cSCC cells allows the cells to obtain transcriptionally distinct cell signatures. This method allows us to recapitulate changes that occur in the early stages of cancer progression, as well as perform future epigenetic and sequencing studies. Moreover, our RDEB-cSCC-iKC tumor model offers a valuable tool for identifying potential biomarkers of tumorigenesis and possible therapeutic targets for cSCC. In the future, we hope to further study the roles of *SOD2* and *FEN1* in the initiation of RDEB-cSCC, as well as develop therapies to target specific genes within the clusters. Taken together, the manipulation of iPSC technology and transcriptomic analyses create a powerful tool for understanding the changes that occur during cSCC tumorigenesis and progression.

## 4. Materials and Methods

### 4.1. Cell Culture

Primary human fibroblasts and keratinocytes were obtained from skin biopsies of RDEB patients and healthy control patients. The RDEB-SCC cell lines used in this study were kindly provided to us from Dr Andrew South. Fibroblasts were grown in Dulbecco’s modified Eagle’s medium (DMEM) (Gibco, Gaithersburg, MD, USA) supplemented with 10% FBS (Gibco, Gaithersburg, MD, USA) and 1% penicillin-streptomycin (Invitrogen, Carlsbad, CA, USA). Keratinocytes were grown in EpiLife medium (Gibco, Gaithersburg, MD, USA) supplemented with 1% penicillin-streptomycin (Invitrogen, Carlsbad, CA, USA) and all SCC cells were grown in Human Keratinocyte Growth Serum (HKGS) medium (Gibco, Gaithersburg, MD, USA) supplemented with 1% penicillin-streptomycin (Invitrogen, Carlsbad, CA, USA). The cells were grown at 37 °C and kept at 5% CO_2_. Media was changed every other day.

### 4.2. Episomal Plasmid-Based Reprogramming Method to Generate RDEB-cSCC-iPSCs

Integration-free iPSCs from RDEB-SCC cell lines were generated as previously described. Briefly, 2 × 10^6^/mL RDEB-cSCCs were seeded into 10 cm^2^ culture dishes and used for electroporation. RDEB-cSCC were cultured in EpiLife medium. On day 0 and 3, cells were electroporated with four reprogramming vectors (pCXLE-hOCT3/4-shp53-F: OCT3/4 and p53 shRNA, pCXLE-hSK: SOX2 and KLF4, pCXLE-hUL: _L_-Myc and LIN28, pCXWB-EBNA1: EBNA1). At day 15, the cells showed morphological changes indicative of reprogramming, and were plated after day 25 on vitronectin-coated plates under feeder-free conditions. The cells were cultured with Essential 8 medium (Gibco, Gaithersburg, MD, USA) for iPSC cells. Gentle cell dissociation reagent (StemCell Technology, Vancouver, WA, Canada), an enzyme-free reagent, was used for the dissociation of iPSCs into cell aggregates for routine passaging and expansion. To confirm their pluripotency, the iPSCs were analyzed for the expression of stem cells markers and differentiation capacity.

### 4.3. Human STEMCCa Lentivirus-Based Reprogramming Method to Generate RDEB-cSCC-iPSCs

The Human Stem Cell Cassette (STEMCCA) lentivirus reprogramming vector is comprised of the humanized transcription factors Oct-4, Klf4, SOX-2, and c-Myc (OKSM), separated by the self-cleaving 2A peptide and internal ribosome entry site (IRES) sequences, driven by the EF-1alpha constitutive promoter. The Cre/LoxP-regulated version has flanking LoxP sites incorporated into the vector, which we used for the infection of the RDEB-cSCC cells. The infection was performed according to manufacturer’s protocol.

### 4.4. Feeder-Free Direct Differentiation of SCC iPSCs to SCC-Keratinocytes (iKC)

Feeder-free iPSC-derived keratinocytes (SCC iKC) were generated with clinical applications in mind. iPSCs were maintained in E8 medium without a feeder layer. For keratinocyte differentiation, small clumps of iPSCs were sub-cultured onto vitronectin-coated plates. At 70% confluence, the iPSCs were harvested to form embryoid bodies (EB). Following EB formation, the EBs were transferred to 10 cm^2^ culture dishes and were incubated in E6 medium supplemented with 1 μM all-trans retinoic acid (RA)(Sigma-Aldrich, St. Louis, MO, USA) and 10 ng/mL BMP4 (R&D Systems, Minneapolis, MN, USA) for 7 days. At day 7, the medium was switched to Defined Keratinocyte Serum-Free Medium (Gibco, Gaithersburg, MD, USA), and iPSCs were maintained in culture without passaging for 12, 30, or 60 days. After 4 passages, the iPSC-derived keratinocytes were subjected to in vitro and in vivo functional analyses.

### 4.5. MTS Assay

The RDEB-cSCC and RDEB-cSCC-iKC were seeded in triplicates and the absorbance was read at 490 nm using the CellTiter 96^®^ AQueous Assay (Promega #TB169, Madison, WI, USA). The CellTiter 96^®^ AQueous Non-Radioactive Cell Proliferation Assay(a) is a colorimetric method for determining the number of viable cells in proliferation or chemosensitivity assays. The CellTiter 96^®^ AQueous Assay is composed of solutions of a novel tetrazolium compound [3-(4,5-dimethylthiazol-2-yl)-5-(3-carboxymethoxyphenyl)-2-(4-sulfophenyl)-2H-tetrazolium, inner salt; MTS(a)] and an electron coupling reagent (phenazine methosulfate; PMS). MTS is bioreduced by cells into a formazan product that is soluble in tissue culture medium. The absorbance of the formazan at 490 nm can be measured directly from 96-well assay plates without additional processing. The conversion of MTS into aqueous, soluble formazan is accomplished by dehydrogenase enzymes found in metabolically active cells. The quantity of formazan product as measured by the amount of 490 nm absorbance is directly proportional to the number of living cells in culture.

### 4.6. Generation of 3D Human Skin Equivalents

3D human skin equivalents (HSEs) were generated as previously described, using primary fibroblasts and iKCs [20]. After maintaining the HSEs in epidermalization medium for 7 days, they were transferred to a new plate and exposed to the air-liquid interface in cornification medium. After 7 days in cornification medium, they were harvested for grafting onto mice.

### 4.7. Engraftment of 3D Skin Constructs on Mice

All experimental animal protocols were approved by the Institutional Animal Care and Use Committee at Columbia University, and the grafting was performed as previously reported [33]. Briefly, 0.8 cm of skin was removed from the dorsal anterior-posterior midline surface of 8–10-week-old NU(NCR)-Fon1nu nude mice (Charles River, Wilmington, MA, USA) using the pinch-cutting technique. The 3D skin constructs were placed on the recipient and secured with 4 sutures (7-0 Nylon) in a simple interrupted pattern around the edge of the graft. To protect the skin equivalent, we adapted a method described for the establishment of humanized skin grafts [33], whereby the piece of skin initially removed from the mouse was freeze-thawed 3 times by placing in DMEM, then inserted into liquid nitrogen prior to warming in a beaker of water. This decellularized and devitalized skin was placed on top of the 3D skin construct on the mouse and was sutured in place with interrupted sutures. A OpSite Flexifix Transparent Film (Smith & Nephew, Andover, MA, USA) bandage was then wrapped around the mouse to hold the graft in place. The mice were euthanized after 2 months for analysis.

### 4.8. Bioluminescence Analysis

The In Vivo Imaging System (IVIS) Optical Imaging system was used to detect the bioluminescence signal in the skin grafts made from RDEB-cSCC expressing the luciferase gene. Mice were anesthetized with isoflurane before imaging. Luciferin was administered to the mouse to induce bioluminescence. Once the imaging was complete, mice were kept homeothermic via a warming pad and monitored for recuperation from the isoflurane anesthesia until fully awake and oriented [21]. Following image acquisition, total flux and average radiance was quantified using the Living Image software.

### 4.9. Immunostaining and Imaging

For immunostaining analyses, cells were fixed in chamber slides with 4% paraformaldehyde. Samples were incubated with primary antibodies overnight at 4 °C. The following primary antibodies were used for IF experiments: SOX2 (48-1400 Invitrogen, Carlsbad, CA, USA), NANOG (4903 Cell Signaling Technology, Danvers, MA, USA), TRA-1-60 (MA1-023 Invitrogen, Carlsbad, CA, USA), TRA-1-81 (MA1-024 Invitrogen, Carlsbad, CA, USA). After washing with phosphate-buffered saline, samples were incubated with fluorophore-conjugated secondary antibodies (Invitrogen, Carlsbad, CA, USA) for 2 h at room temperature. Slides were covered with coverslips using VECTASHIELD mounting medium containing 4′,6-diamidino-2-phenylindole (DAPI) (Vector Laboratories, San Francisco, CA, USA) and examined using a Zeiss LSM 5 Exciter confocal laser scanning microscope.

### 4.10. Transcriptomic Analysis

The total RNA was isolated from RDEB-cSCC, RDEB-cSCC-iPSCs and RDEB-cSCC-iKCs according to the manufacturer’s protocol (Qiagen, Hilden, Germany) and RNA-seq was performed at Genewiz, NJ, USA. RNA-Sequencing libraries were prepared by TruSeq Stranded total RNA Library Prep (Illumina Corp., San Diego, CA, USA) according to the manufacturer’s standard protocol. Libraries were sequenced on the HiSeq instrument (Illumina Co., San Diego, CA, USA) following the manufacturer’s protocol. FASTQ-formatted sequences were analyzed for quality control by the FASTQC open-source software. Trimmomatic tool was used to trim raw sequences. Sequences with quality score below 24 were excluded. Then HISAT2 software was used to align reads to the reference genome sequence (hg38). Transcript expression was comprehensively quantified with Cufflinks software, which performed: transcript assembly, quantification of expression as FPKM (fragments per kilobase of transcript per million mapped reads) values, and normalization. Normalized FPKM values were used for downstream analysis steps. Exploratory data analysis was used to cluster the gene expression data. The rank candidate genes were typed from each cluster based on functional similarity to training gene list using machine learning algorithm (ToppGene tool). This list was obtained from publication Cho et al. 2018 (ToppGene tool) [4]. To determine the biological functional processes of the candidate genes, the online tool provided in the DAVID v. 6.8 (Database for Annotation, Visualization, and Integrated Discovery) website was used. A *p*-value of < 0.05 was used to indicate statistical significance.

### 4.11. Statistical Analysis

Differential expression analyses were carried out using the non-parametric Wilcoxon rank-sum test comparing the samples and normal human control cells. Significant genes were detected based on the stringent criteria of *p* values < 0.05 with a fold change greater than 2. Data were compared between groups using two-tailed unpaired t tests. Statistical significance was set at * *p* < 0.05.

## Figures and Tables

**Figure 1 ijms-22-00245-f001:**
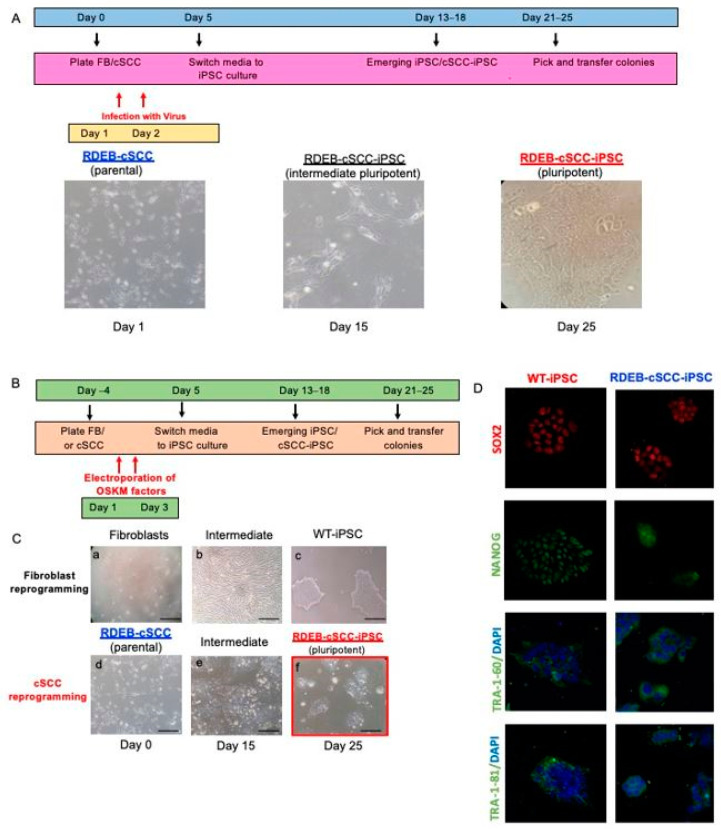
Reprogramming and characterization of reprogrammed RDEB-cSCC cells. (**A**) Timeline for lentivirus-based method to reprogram RDEB-cSCC into RDEB-cSCC-iPSC. Representative pictures taken throughout the reprogramming method are also shown. (**B**) Timeline for feeder-free method to reprogram RDEB-cSCC into RDEB-cSCC-iPSC using an episomal plasmid. (**C**) Morphology of RDEB-cSCC-iPSC derived from RDEB-cSCC compared to WT-iPSC derived from fibroblasts. iPSCs appear to be morphologically similar (small cell shape, large nuclear-to-cytoplasmic ratio, prominent nucleoli, tight and flat colonies). (**D**) Immunofluorescence staining of stem cells markers in WT-iPSCs and RDEB-cSCC-iPSCs.

**Figure 2 ijms-22-00245-f002:**
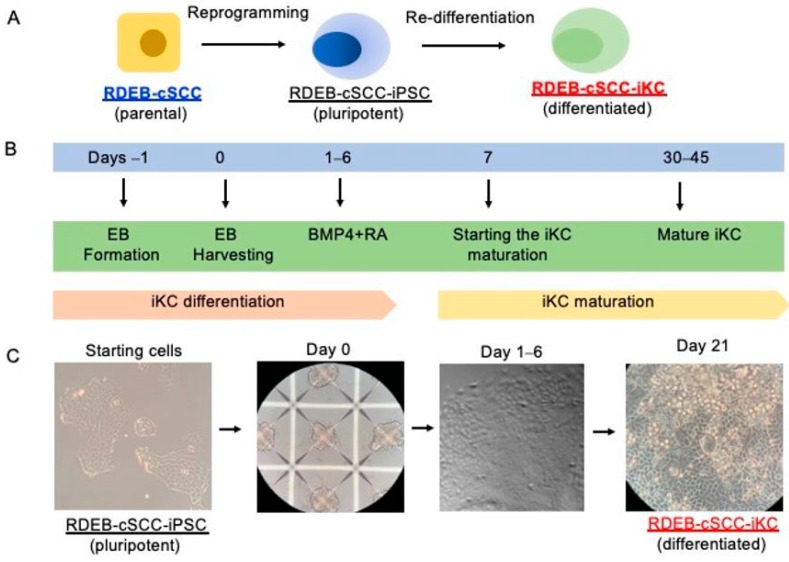
Re-differentiation of RDEB-cSCC into RDEB-cSCC-iKCs. (**A**) General schematic for reprogramming and re-differentiation methods. (**B**) Timeline for method to differentiate RDEB-cSCC-iPSC into RDEB-cSCC-iKC. (**C**) Representative pictures of the differentiation of RDEB-cSCC-iPSC into RDEB-cSCC-iKC.

**Figure 3 ijms-22-00245-f003:**
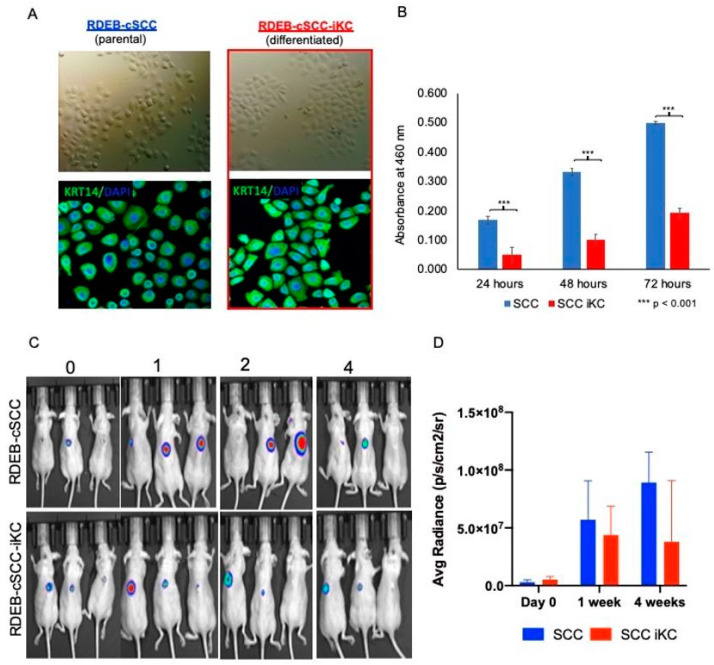
In vitro and in vivo characterization of RDEB-cSCC-iKC. (**A**) Representative pictures showing the morphological differences between parental RDEB-SCC and re-differentiated RDEB-cSCC-iKC. Keratin 14 (KRT14) is shown in green and nuclei are stained in blue (DAPI) on the representative immunofluorescence pictures. (**B**) RDEB-cSCC and RDEB-cSCC-iKC cell numbers over 72 h on absorbance at 490 nm measured using the CellTiter 96^®^ AQueous Assay. *** *p* < 0.001 (**C**) Bioluminescence signal intensities over a 4-week course for tumor models derived from RDEB-cSCC and RDEB-cSCC-iKC. (**D**) Average radiance over a 4-week course for RDEB-cSCC and RDEB-cSCC-iKC tumor models.

**Figure 4 ijms-22-00245-f004:**
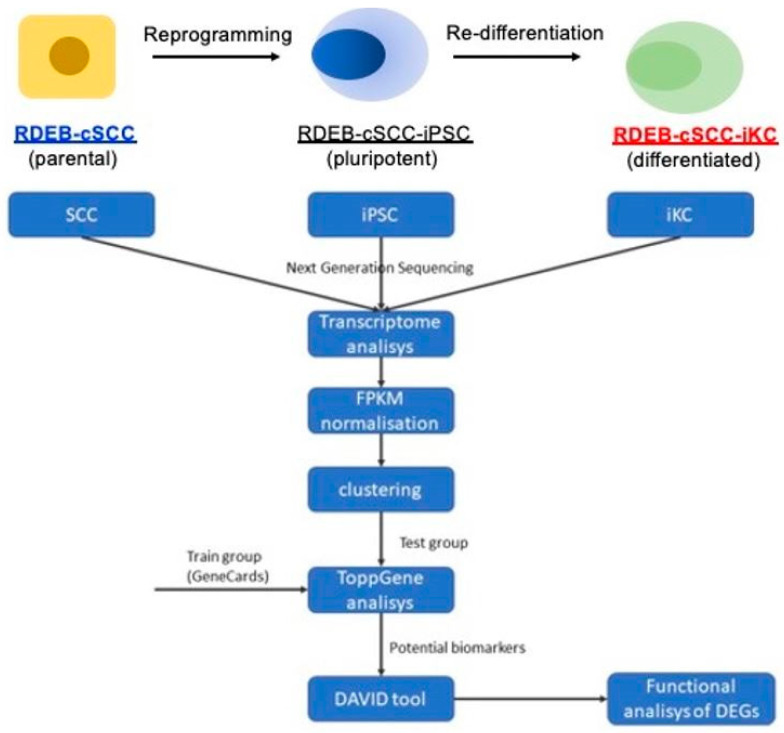
Schematic of bioinformatics analysis. General schematic demonstrating methods for RNA-seq analysis and comparison of gene expression profiles.

**Figure 5 ijms-22-00245-f005:**
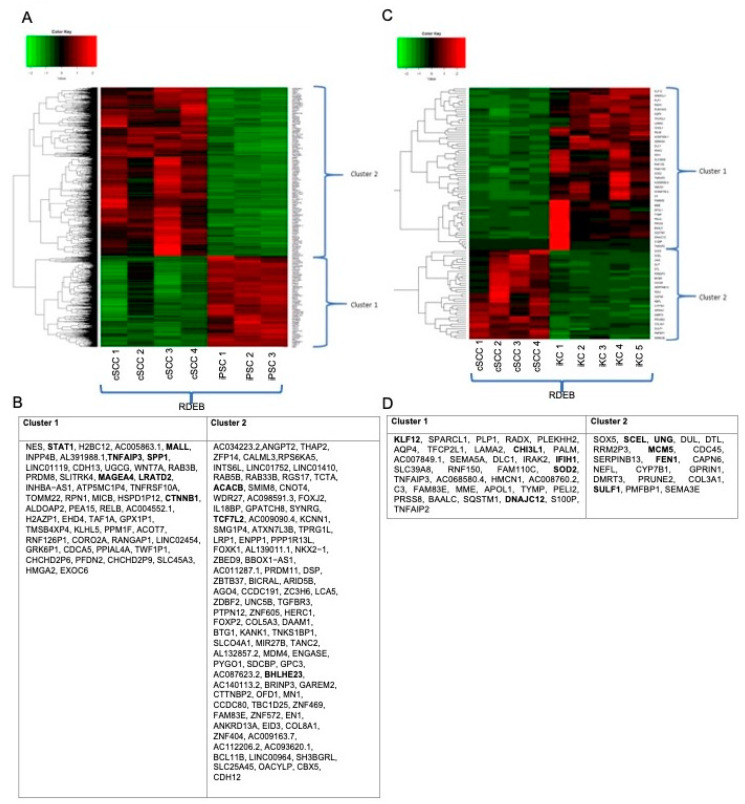
Differential expression analyses. (**A**) Two clusters revealed by differential expression analyses, with opposite expression levels. Left: Differential expression analysis between RDEB-cSCC and RDEB-cSCC-iPSC. (**B**)The first cluster includes genes with higher expression in RDEB-cSCC-iPSC, while the second includes genes with lower expression. (**C**) Right: Differential expression analysis between RDEB-cSCC and RDEB-cSCC-iKC. (**D**) The first cluster includes genes with higher expression in RDEB-cSCC-iKC, while the second includes genes with lower expression.

## Data Availability

Not applicable.

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
