# Peer review of "Reprogramming and Differentiation of Cutaneous Squamous Cell Carcinoma Cells in Recessive Dystrophic Epidermolysis Bullosa"

_ijms, 2020, doi:10.3390/ijms22010245_

Round 1

Reviewer 1 Report

In the presented manuscript, the authors approached a fundamental question in the field why the RDEB-cSCCs are tremendously aggressive. To investigate this, the authors developed a model using iPSC technology. The authors generated iPS cells from RDEB cSCC, and further differentiated the RDEB-cSCC-iPSC to RDEB-cSCC-iKC. In addition, the authors analyzed the gene expression signatures of RDEB-cSCC, RDEB-cSCC-iPSC, and RDEB-cSCC-iKC, and identified the genes specifically expressed in each cell. Although the reprograming and differentiation techniques using RDEB-cSCC were nicely shown, the review has concerns on the scientific hypothesis and the interpretations of results.

Major points:

  • To the reviewer, it was unclear exactly which process in tumorigenesis/tumor development the system models. As nicely described in the previous paper (Sci Transl Med . 2018 Aug 22;10(455):eaas9668.), same as other tumors, RDEB-cSCCs have enormous number of mutations on their genomes (at least over 100 nonsynonymous mutations). Thus, RDEB-cSCC-iKCs inevitably inherit the same mutations original RDEB-cSCCs have. The keratinocytes in RDEB patients should not have this number of mutations; most of the mutations in RDEB-cSCCs is acquired in the course of tumor development. RDEB-cSCC-iKC established in this study may look like keratinocytes, however the cells have the same number of mutations the parental cSCC cells have, which cannot be happened in the real world. The authors should describe exactly how the system presented here can be used in the tumor studies.
  • The authors reprogrammed and re-differentiated RDEB-cSCC using iPSC technology. To the reviewer, it was unclear how many cSCC with different origins they used in this study. To generalize their findings, the authors should provide additional examples showing the system can be applied to other RDEB-cSCC too.
  • It was unclear to the reviewer whether this system really recapitulates the tumor development in RDEB. To test this, the authors should obtain SCC from RDEB-cSCC-iKC and compare their gene expression signatures with the original RDEB-cSCC.

Minor points:

  • Figure 1A cannot be read due to low resolution.
  • Figure 3C needs quantification.
  • Inconsistency in labeling. iPSc or iPSCC, RDEB-cSCC or RDEB-SCC. 

Author Response

Manuscript ID-ijms-1026608

Point-by-point responses to reviewer comments

Reviewer #1

Comments and suggestions to the Author:In the presented manuscript, the authors approached a fundamental question in the field why the RDEB-cSCCs are tremendously aggressive. To investigate this, the authors developed a model using iPSC technology. The authors generated iPS cells from RDEB cSCC, and further differentiated the RDEB-cSCC-iPSC to RDEB-cSCC-iKC. In addition, the authors analyzed the gene expression signatures of RDEB-cSCC, RDEB-cSCC-iPSC, and RDEB-cSCC-iKC, and identified the genes specifically expressed in each cell. Although the reprograming and differentiation techniques using RDEB-cSCC were nicely shown, the review has concerns on the scientific hypothesis and the interpretations of results.

Answer:We thank the Reviewer for the positive comments and appreciation of our work on model using iPSC technology to study RDEB cSCC in vitroand in vivo. We addressed all the Reviewers’ comments and have also provided new quantification data that significantly improved our manuscript.

Major points:

      Question 1:  To the reviewer, it was unclear exactly which process in tumorigenesis/tumor         development the system models. As nicely described in the previous paper (Sci Transl Med.           2018 Aug 22;10(455):eaas9668.), same as other tumors, RDEB-cSCCs have enormous number      of mutations on their genomes (at least over 100 nonsynonymous mutations). Thus, RDEB-        cSCC-iKCs inevitably inherit the same mutations original RDEB-cSCCs have. The keratinocytes          in RDEB patients should not have this number of mutations; most of the mutations in RDEB-       cSCCs is acquired in the course of tumor development. RDEB-cSCC-iKC established in this        study may look like keratinocytes, however the cells have the same number of mutations the         parental cSCC cells have, which cannot be happened in the real world. The authors should       describe exactly how the system presented here can be used in the tumor studies.

Answer: Thank you reviewer for your comments. We agree with the reviewer that the process of tumorigenesis is driven by specific gene expression patterns which are influenced by mutation signatures but can also be influenced by the epigenetic state of the cell. Epigenetic changes are mitotically and/or meiotically heritable changes in gene function that cannot be explained by changes in the DNA sequences but lead to the propagation of heritable changes in phenotype (Cavalli & Heard, 2019). There are many studies supporting this hypothesis. DNA methylation and histone modifications are forms of epigenetic modifications that have profound impacts on the epigenetic landscape, gene expression and cell behavior (Wainwright & Scaffidi, 2017). Studies have shown that the promotion of cancer initiation and development is accompanied by various epigenetic changes (Jones & Laird, 1999). Recently, studies have identified that the aberrant methylation of genes was correlated with the progression of squamous cell carcinoma, corroborating the belief that epigenetic alterations play a significant role in the initiation and development of this disease (Hervas-Marin et al., 2019).

Cellular reprogramming technology can be utilized to further understand epigenetics-driven cancer development (Yamada, Haga, & Yamada, 2014). Previous papers have highlighted how reprogramming technology can be used to modify the epigenome without affecting the underlying genomic sequences to study the onset of cancer (Khoshchehreh et al., 2019). Thus, in our study, we hypothesized that using our reprogramming and re-differentiation model, we can study epigenetic alterations without modifying the underlying genomic sequences, providing a rationale behind why the RDEB-cSCC-iKC have the same number of mutations as the parental cSCC cells. The new comments are incorporated into the introduction in the new version of our manuscript lines 60-65.  

      Question 2:The authors reprogrammed and re-differentiated RDEB-cSCC using iPSC    technology. To the reviewer, it was unclear how many cSCC with different origins they used in   this study. To generalize their findings, the authors should provide additional examples showing   the system can be applied to other RDEB-cSCC too.

  Answer:Thank you reviewer for your comments. For this study, the authors used 4 clones of RDEB-cSCCs to demonstrate the reproducibility of this concept. We are currently working on reprogramming and differentiating multiple primary RDEB-cSCC to further generalize our findings.

      Question 3:It was unclear to the reviewer whether this system really recapitulates the tumor     development in RDEB. To test this, the authors should obtain SCC from RDEB-cSCC-iKC and    compare their gene expression signatures with the original RDEB-cSCC.

Answer: Thank you reviewer for your comments. For the scope of this study, we only planned to differentiate RDEB-cSCC into RDEB-cSCC-iKC to study SCC; however, this is an excellent idea to obtain SCC from RDEB-cSCC-iKC and compare their gene expression signature with the orginal RDEB-cSCC, and we will consider this for future studies. We have previously reported fibroblast and keratinocytes differentiation from iPSC(Jackow et al., 2019). Thus, we will use this protocol as a starting point for future studies on other differentiation pathways from RDEB-cSCC-iKC into SCC.

Minor points:

  • Figure 1A cannot be read due to low resolution.

      Answer: The new picture is incorporated in Figure 1A.

  • Figure 3C needs quantification.

      Answer: The new quantification data are presented in Figure 3D.

  • Inconsistency in labeling. iPSc or iPSCC, RDEB-cSCC or RDEB-SCC. 

      Answer: The labeling is consistent throughout the new version of our manuscript.

Answer:The minor issues pointed by reviewer are corrected and included in the revised version of our manuscript.

References

Cavalli, G., & Heard, E. (2019). Advances in epigenetics link genetics to the environment and disease. Nature, 571(7766), 489-499. doi:10.1038/s41586-019-1411-0

Hervas-Marin, D., Higgins, F., Sanmartin, O., Lopez-Guerrero, J. A., Bano, M. C., Igual, J. C., . . . Sandoval, J. (2019). Genome wide DNA methylation profiling identifies specific epigenetic features in high-risk cutaneous squamous cell carcinoma. PLoS One, 14(12), e0223341. doi:10.1371/journal.pone.0223341

Jackow, J., Guo, Z., Hansen, C., Abaci, H. E., Doucet, Y. S., Shin, J. U., . . . Christiano, A. M. (2019). CRISPR/Cas9-based targeted genome editing for correction of recessive dystrophic epidermolysis bullosa using iPS cells. Proc Natl Acad Sci U S A. doi:10.1073/pnas.1907081116

Jones, P. A., & Laird, P. W. (1999). Cancer epigenetics comes of age. Nat Genet, 21(2), 163-167. doi:10.1038/5947

Wainwright, E. N., & Scaffidi, P. (2017). Epigenetics and Cancer Stem Cells: Unleashing, Hijacking, and Restricting Cellular Plasticity. Trends Cancer, 3(5), 372-386. doi:10.1016/j.trecan.2017.04.004

Yamada, Y., Haga, H., & Yamada, Y. (2014). Concise review: dedifferentiation meets cancer development: proof of concept for epigenetic cancer. Stem Cells Transl Med, 3(10), 1182-1187. doi:10.5966/sctm.2014-0090

Reviewer 2 Report

A very novel study with excellent data and data representation. I would ask the authors to consider commenting on the following: Were time points assessed during the earlier differentiation process either in vitro or in vivo? The authors mention drivers of SCC (line 224) and I am curious if these were assessed at earlier timepoints as well as part of identifying whether there are differential driver impacts. Does this aid in understanding EMT? Can the authors comment on any performed or planned other cell differentiation pathways that were or could be employed (fibroblasts, MSC, etc). Can the authors comment on their findings in relation to the study by Li et al in regards to similar or divergent findings (Multimodal Analysis of Composition and Spatial Architecture in Human Squamous Cell Carcinoma, Cell, Volume 182, Issue 2, 23 July 2020, Pages 497-514.e22) Were the karyotypes of SCC and iPSC SCC maintained? The comparison of normal and RDEBB are more than sufficient but can they offer thoughts on whether rescuing the RDEB SCC with COL7A1 expression has potential to alter cancer progression? This would be important in terms of gene/cell therapy approaches and whether they will impact SCC occurrence/aggressiveness

Author Response

Manuscript ID-ijms-1026608

Point-by-point responses to reviewer comments

Reviewer #2

Comments and suggestions to the Author:A very novel study with excellent data and data representation. I would ask the authors to consider commenting on the following:

Answer: We thank the Reviewer for the positive comments and appreciation of our work on model using iPSC technology to study RDEB cSCC in vitroand in vivo.  We addressed all the Reviewers’ comments below.

Question 1:Were time points assessed during the earlier differentiation process either in vitro or in vivo?

Answer: Thank you reviewer for this question. The entire differentiation process is conducted in vitro, as described in previously published papers (Jackow et al., 2019; Jackow et al., 2020).

Question 2:The authors mention drivers of SCC (line 224 in the first version of the manuscript, in new version of the manuscript it is 240 ) and I am curious if these were assessed at earlier time points as well as part of identifying whether there are differential driver impacts. Does this aid in understanding EMT?

Answer: Thank you reviewer for your comments. Based on a review article published by Condorelli et al., cSCC is initialized by mutations in genes such as TP53, NOTCH1, NOTCH2, CDKN2A, HRAS, and FAT1, all of which are well-known drivers of tumorigenesis. However, factors other than genetics can also contribute to tumor development, such as the microenvironment. Specifically, impaired healing of chronic wounds, inflammation, and intracellular signaling mechanisms, such as the activation of pro-tumorigenic processes like angiogenesis and tumor cell invasion, have been cited to drive cSCC development. In particular, COL7 loss in SCC keratinocytes promotes EMT through various intracellular mechanisms, such as the activation of TGF-β1 signaling (Martins et al., 2009). Through the discovery of novel driver genes and an understanding of the intracellular mechanisms that promote cell migration and invasion, we can better understand EMT and its role in the onset of cSCC.

Question 3:Can the authors comment on any performed or planned other cell differentiation pathways that were or could be employed (fibroblasts, MSC, etc).

Answer: Thank you reviewer for your comments. For the scope of this study, we only employed the differentiation protocol to study SCC; however, this is an excellent idea, and we will consider this for future studies. We have previously reported fibroblast differentiation from iPSC (Jackow et al., 2019). Thus, we will use this protocol as a starting point for future studies on other differentiation pathways.

Question 4:Can the authors comment on their findings in relation to the study by Li et al in regards to similar or divergent findings (Multimodal Analysis of Composition and Spatial Architecture in Human Squamous Cell Carcinoma, Cell, Volume 182, Issue 2, 23 July 2020, Pages 497-514.e22).

Answer:Thank you reviewer for your comments. The study by Ji et al. is an excellentcomplement to our study.  In this study, the researchers have identified four tumor subpopulations within the cSCC. Three of the subpopulations recapitulate normal epidermal states, while one population is composed of tumor-specific keratinocytes (TSK) unique to cancer, which is localized to a fibrovascular niche. Through the integration of single-cell and spatial transcriptomics with single cell resolution, the study showed that TSK cells serve as a hub for intercellular communication. With regards to our study, it will be good to compare the TSK cells subpopulation with our reprogrammed RDEB-cSCC-iKC to validate whether our model is a useful tool to recapture this unique cell subpopulation. This new comment is incorporated into the discussion in the new version of our manuscript line 214-217.

Question 5:Were the karyotypes of SCC and iPSC SCC maintained?

Answer: Thank you reviewer for your comments. We did not analyze the karyotypes of RDEB-SCC and RDEB-iPSC-SCC for this study. However, previously when we established our iPSC reprogramming protocols, we performed karyotype analysis and found no difference between the parental cells and reprogrammed iPSC (Jackow et al., 2019).

Question 6:The comparison of normal and RDEBB are more than sufficient but can they offer thoughts on whether rescuing the RDEB SCC with COL7A1 expression has potential to alter cancer progression? This would be important in terms of gene/cell therapy approaches and whether they will impact SCC occurrence/aggressiveness.

Answer: Thank you reviewer for your comments.  It has been shown in the past that rescuing collagen VII expression in RDEB cells corrects the cell migration and invasion (Nystrom et al., 2013). Gene/cell therapies for RDEB cSCC are currently under the development and, thus, if these therapies could be given to patients prior to tumor development, this will not only improve skin wound healing but will serve as a preventative measure for RDEB-cSCC tumorigenesis. However, after tumor development occurs, treatment is necessary to counteract disease pathogenesis.

References

Jackow, J., Guo, Z., Hansen, C., Abaci, H. E., Doucet, Y. S., Shin, J. U., . . . Christiano, A. M. (2019). CRISPR/Cas9-based targeted genome editing for correction of recessive dystrophic epidermolysis bullosa using iPS cells. Proc Natl Acad Sci U S A. doi:10.1073/pnas.1907081116

Jackow, J., Rami, A., Hayashi, R., Hansen, C., Guo, Z., DeLorenzo, D., . . . Christiano, A. M. (2020). Targeting the JAK/STAT3 pathway with Ruxolitinib in a mouse model of recessive dystrophic epidermolysis bullosa-squamous cell carcinoma. J Invest Dermatol. doi:10.1016/j.jid.2020.08.022

Nystrom, A., Velati, D., Mittapalli, V. R., Fritsch, A., Kern, J. S., & Bruckner-Tuderman, L. (2013). Collagen VII plays a dual role in wound healing. J Clin Invest, 123(8), 3498-3509. doi:10.1172/JCI68127

Round 2

Reviewer 1 Report

The authors provided additional data and explanations in the revised manuscript, however, the important matters the reviewer raised have not fully approached in the revised manuscript.    Point 1 The review agrees that the epigenetic regulation has important roles in tumorigenesis/tumor development. However, the reviewer is still not sure whether the iPSC system described here recapitulates actual tumorigenesis/tumor development processes. First, if the authors propose to study the epigenetic signatures using this system, the authors should demonstrate that the epigenetic signatures are also reprogrammed in the RDEB-cSCC-iPSC and RDEB-cSCC-iKC. Second, the epigenetic changes and genomic mutations are gradually acquired in the course of tumor development. The reviewer is not sure if studying epigenetic regulation in the iKC with cSCC mutations is useful. The epigenetic changes and genomic mutations are NOT independent.      Point 2 The quantification data in Figure3D (why data for week2 is missing?) do not support the description below. Line134. "The tumors formed by parental cells grown faster and were bigger compared to the tumors generated from RDEB-cSCC-iKCs, as shown by the bioluminescence signal intensities."      

Author Response

Point-by-point responses to reviewer comments

Reviewer #1 – Round 2

Comments and suggestions to the Author:The authors provided additional data and explanations in the revised manuscript, however, the important matters the reviewer raised have not fully approached in the revised manuscript.   

Point 1The review agrees that the epigenetic regulation has important roles in tumorigenesis/tumor development. However, the reviewer is still not sure whether the iPSC system described here recapitulates actual tumorigenesis/tumor development processes. First, if the authors propose to study the epigenetic signatures using this system, the authors should demonstrate that the epigenetic signatures are also reprogrammed in the RDEB-cSCC-iPSC and RDEB-cSCC-iKC. Second, the epigenetic changes and genomic mutations are gradually acquired in the course of tumor development. The reviewer is not sure if studying epigenetic regulation in the iKC with cSCC mutations is useful. The epigenetic changes and genomic mutations are NOT independent.     

Answer: Thank you to the reviewer for their comments. Within the scope of this paper, we have not directly studied epigenetic regulation; however, we have indirectly studied these changes through our in vitroand in vivotumorigenicity studies. Through these studies, our group has demonstrated that the reprogrammed and re-differentiated cSCC have a reduced proliferative capacity. Thus, we hypothesize that through the manipulation of iPSC technology, we can convert cancer cells to represent an earlier stage in the evolution of aggressive cSCC. One study by Kim et al.highlights how the reprogramming of somatic cells reverses the process of cell specification through epigenetic modifications, allowing for the erasure of tissue-specific DNA methylation and the re-establishment of the embryonic methylation patterns [1]. Additionally, other studies have elucidated the use of iPSC models to study early changes in cancer. These studies have demonstrated the value of utilizing reprogramming technology to model disease pathogenesis for colon cancer, osteosarcoma, and gastrointestinal cancer [2]–[4]. The new comments are incorporated into the new version of our manuscript lines 189-195. 

                       The aim of this paper is to represent a model that can be used to study epigenetic changes. In further studies, we plan to examine the epigenetic signatures of the parental, reprogrammed, and re-differentiated cells. We agree with the reviewer that epigenetic changes and genomic mutations are not independent of one another. However, we believe studying epigenetic regulation in the iKC with cSCC mutations is useful as epigenetic changes have been previously reported to drive gene expression. A recent study by Atlasi et al.highlighted how differential gene expression can be driven by enhancer activation and chromatin accessibility, delineating the role of epigenetic modulation in driving changes in gene expression, independent of mutational changes [5]. Additionally, studying epigenetic changes can be useful when developing targeted therapies for cancer. A study by Miyoshi et al. demonstrated how tumor-suppressor gene P16was repressed in induced pluripotent cancer cells, highlighting a potential target for gastrointestinal cancer treatment [4]. The new comments are incorporated into the new version of our manuscript lines 243-253. 

Point 2The quantification data in Figure3D (why data for week2 is missing?) do not support the description below. Line134. "The tumors formed by parental cells grown faster and were bigger compared to the tumors generated from RDEB-cSCC-iKCs, as shown by the bioluminescence signal intensities."      

Answer: Thank you to the reviewer for their comments. The updated data (now including week 2) in Figure 3D show that the bioluminescence signal intensities are higher for parental cSCC than iKCs, supporting the statement in line 140.

[1]       K. Kim et al., “Donor cell type can influence the epigenome and differentiation potential of human induced pluripotent stem cells,” Nat. Biotechnol., 2011, doi: 10.1038/nbt.2052.

[2]       N. Oshima et al., “Induction of cancer stem cell properties in colon cancer cells by defined factors,” PLoS One, 2014, doi: 10.1371/journal.pone.0101735.

[3]       D. F. Lee et al., “Modeling familial cancer with induced pluripotent stem cells,” Cell, 2015, doi: 10.1016/j.cell.2015.02.045.

[4]       N. Miyoshi et al., “Defined factors induce reprogramming of gastrointestinal cancer cells,” Proc. Natl. Acad. Sci. U. S. A., 2010, doi: 10.1073/pnas.0912407107.

[5]       Y. Atlasi et al., “Epigenetic modulation of a hardwired 3D chromatin landscape in two naive states of pluripotency,” Nat. Cell Biol., 2019, doi: 10.1038/s41556-019-0310-9.
